# Language Is Not All You Need: Aligning Perception with Language Models

Shaohan Huang,* Li Dong,* Wenhui Wang,* Yaru Hao,* Saksham Singhal,* Shuming Ma*
Tengchao Lv, Lei Cui, Owais Khan Mohammed, Barun Patra, Qiang Liu, Kriti Aggarwal
Zewen Chi, Johan Bjorck, Vishrav Chaudhary, Subhojit Som, Xia Song, Furu Wei[†]
Microsoft
https://github.com/microsoft/unilm

## Abstract

A big convergence of language, multimodal perception, action, and world modeling is a key step toward artificial general intelligence. In this work, we introduce KOSMOS-1, a Multimodal Large Language Model (MLLM) that can perceive general modalities, learn in context (i.e., few-shot), and follow instructions (i.e., zero-shot). Specifically, we train KOSMOS-1 from scratch on web-scale multimodal corpora, including arbitrarily interleaved text and images, image-caption pairs, and text data. We evaluate various settings, including zero-shot, few-shot, and multimodal chain-of-thought prompting, on a wide range of tasks without any gradient updates or finetuning. Experimental results show that KOSMOS-1 achieves impressive performance on (i) language understanding, generation, and even OCR-free NLP (directly fed with document images), (ii) perception-language tasks, including multimodal dialogue, image captioning, visual question answering, and (iii) vision tasks, such as image recognition with descriptions (specifying classification via text instructions). We also show that MLLMs can benefit from cross-modal transfer, i.e., transfer knowledge from language to multimodal, and from multimodal to language. In addition, we introduce a dataset of Raven IQ test, which diagnoses the nonverbal reasoning capability of MLLMs.

## 1 Introduction: From LLMs to MLLMs

Large language models (LLMs) have successfully served as a general-purpose interface across various natural language tasks [1]. The LLM-based interface can be adapted to a task as long as we are able to transform the input and output into texts. For example, the input of the summarization task is a document and the output is its summary. So we can feed the input document into the language model and then produce the generated summary.

Despite the successful applications in natural language processing, it is still struggling to natively use LLMs for multimodal data, such as image, and audio. Being a basic part of intelligence, multimodal perception is a necessity to achieve artificial general intelligence, in terms of knowledge acquisition and grounding to the real world. More importantly, unlocking multimodal input [2, 3, 4, 5, 6, 7] greatly widens the applications of language models to more high-value areas, such as multimodal machine learning, document intelligence, and robotics.

In this work, we introduce KOSMOS-1, a Multimodal Large Language Model (MLLM) that can perceive general modalities, follow instructions (i.e., zero-shot learning), and learn in context (i.e., few-shot learning). The goal is to align perception with LLMs, so that the models are able to see and talk. To be specific, we follow METALM [3] to train the KOSMOS-1 model from scratch. As shown in Figure 1, a Transformer-based language model is regarded as the general-purpose interface, and perception modules are docked with the language model. We train the model on web-scale multimodal

---

* Equal contribution. † Corresponding author.

37th Conference on Neural Information Processing Systems (NeurIPS 2023).

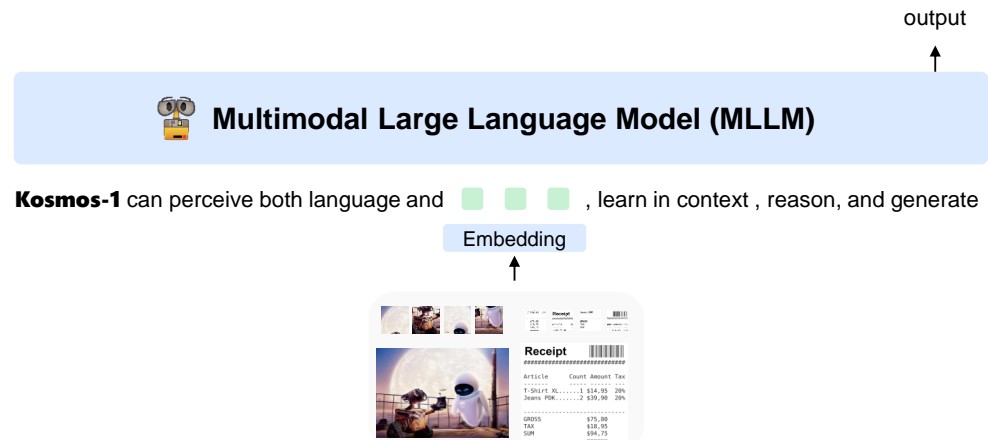

Figure 1: KOSMOS-1 is a multimodal large language model (MLLM) that is capable of perceiving multimodal input, following instructions, and performing in-context learning for not only language tasks but also multimodal tasks. In this work, we align vision with large language models (LLMs), advancing the trend of going from LLMs to MLLMs.

corpora, i.e., text data, arbitrarily interleaved images and texts, and image-caption pairs. In addition, we calibrate the instruction-following capability across modalities by transferring language-only data.

The KOSMOS-1 model natively supports language, perception-language, and vision tasks. In addition to various natural language tasks, the KOSMOS-1 models natively handle a wide range of perception-intensive tasks, spanning visual dialogue, visual explanation, visual question answering, image captioning, simple math equation, OCR, and zero-shot image classification with descriptions. We also build an IQ test benchmark following Raven's Progressive Matrices [8, 9], which evaluates the capability of nonverbal reasoning for MLLMs. The examples show that the native support of multimodal perception enables new opportunities to apply LLMs to new tasks. Moreover, we show that MLLMs achieve better commonsense reasoning performance compared with LLMs, which indicates cross-modal transfer helps knowledge acquisition.

The key takeaways are as follows:

**From LLMs to MLLMs.** Properly handling perception is a necessary step toward artificial general intelligence. The capability of perceiving multimodal input is critical to LLMs. First, multimodal perception enables LLMs to acquire commonsense knowledge beyond text descriptions. Second, aligning perception with LLMs opens the door to new tasks, such as robotics, and document intelligence. Third, the capability of perception unifies various APIs, as graphical user interfaces are the most natural and unified way to interact with. For example, MLLMs can directly read the screen or extract numbers from receipts. We train the KOSMOS-1 models on web-scale multimodal corpora, which ensures that the model robustly learns from diverse sources. We not only use a large-scale text corpus but also mine high-quality image-caption pairs and arbitrarily interleaved image and text documents from the web.

**Language models as general-purpose interfaces.** Following the philosophy proposed in METALM [3], we regard language models as a universal task layer. Because of the open-ended output space, we are able to unify various task predictions as texts. Moreover, natural-language instructions and action sequences (such as programming language) can be well handled by language models. LLMs also serve as basic reasoners [10], which is complementary to perception modules on complex tasks. So it is natural to align world, action, and multimodal perception with the general-purpose interface, i.e., language models.

**New capabilities of MLLMs.** Apart from the capabilities found in previous LLMs [1, 11], MLLMs enable new usages and possibilities. First, we can conduct zero- and few-shot multimodal learning by using natural language instructions and demonstration examples. Second, we observe promising signals of nonverbal reasoning by evaluating the Raven IQ test, which measures the fluid reasoning

ability of humans. Third, MLLMs naturally support multi-turn interactions for general modalities, such as multimodal dialogue.

## 2 KOSMOS-1: A Multimodal Large Language Model

KOSMOS-1 is a multimodal language model that can perceive general modalities, follow instructions, learn in context, and generate outputs. Given the previous context, the model learns to generate texts in an auto-regressive manner. Specifically, the backbone of KOSMOS-1 is a Transformer-based causal language model. Apart from text, other modalities are embedded and fed into the language model. The Transformer decoder serves as a general-purpose interface to multimodal input. We train KOSMOS-1 on multimodal corpora, including monomodal data, cross-modal paired data, and interleaved multimodal data. Once the models are trained, we can directly evaluate the models in zero-shot and few-shot settings on both language tasks and multimodal tasks.

### 2.1 Input Representation

The Transformer decoder perceives general modalities in a unified way. For input format, we flatten input as a sequence decorated with special tokens. Specifically, we use  and  to denote start- and end-of-sequence. The special tokens <image> and </image> indicate the beginning and end of encoded image embeddings. For example, " *document* " is a text input, and " *paragraph* <image> Image Embedding </image> *paragraph* " is an interleaved image-text input.

An embedding module is used to encode both text tokens and other input modalities into vectors. Then the embeddings are fed into the decoder. For text tokens, we use a lookup table to map them into embeddings. For the modalities of continuous signals (e.g., image, and audio), it is also feasible to represent inputs as discrete code and then regard them as "foreign languages" [4, 12]. In this work, following [3], we employ a vision encoder as the embedding module for input images. In addition, Resampler [5] is used as an attentive pooling mechanism to reduce the number of image embeddings.

### 2.2 Multimodal Large Language Models (MLLMs)

After obtaining the embeddings of an input sequence, we feed them into the Transformer-based decoder. The left-to-right causal model processes the sequence in an auto-regressive manner, which produces the next token by conditioning on past timesteps. The causal masking is used to mask out future information. A softmax classifier upon Transformer is used to generate tokens over the vocabulary.

MLLMs serve as general-purpose interfaces [3] that can perform interactions with both natural language and multimodal input. The framework is flexible to handle various data types, as long as we can represent input as vectors. MLLMs combine the best of two worlds. First, the language models naturally inherit the capabilities of in-context learning and instruction following. Second, perception is aligned with language models by training on multimodal corpora.

The implementation is based on the library TorchScale [13], which is designed for large-scale model training. Compared with the standard Transformer architecture, we include the following modifications: We use MAGNETO [14], a Transformer variant, as the backbone architecture and XPOS [15] relative position encoding for better long-context modeling.

### 2.3 Multimodal Training Data

The models are trained on web-scale multimodal corpora. The training datasets consist of text corpora, image-caption pairs, and interleaved data of images and texts.

**Text Corpora**   We train our model with The Pile [16] and Common Crawl (CC). The Pile is a massive English text dataset built for training large-scale language models. We exclude data splits from GitHub, arXiv, Stack Exchange, and PubMed Central. We also include the Common Crawl snapshots (2020-50 and 2021-04) datasets, CC-Stories, and RealNews datasets [17, 18].

**Image-Caption Pairs**   The image-caption pairs are constructed from several datasets, including English LAION-2B [19], LAION-400M [20], COYO-700M [21], and Conceptual Captions [22, 23].

English LAION-2B, LAION-400M, and COYO-700M are collected from web pages of the Common Crawl web data by extracting image sources and the corresponding alt-text. Conceptual Captions are also from internet web pages.

**Interleaved Image-Text Data** We collect interleaved multimodal data from the Common Crawl snapshot, which is a publicly available archive of web pages. We use a filtering process to select about 71 millions web pages from the original 2 billions web pages in the snapshot. We then extract the text and images from the HTML of each selected web page.

## 2.4 Training Objective

The KOSMOS-1 training is conducted on web-scale multimodal corpora, including monomodal data (e.g., text corpus), cross-modal paired data (e.g., image-caption pairs), and interleaved multimodal data (e.g., documents of arbitrarily interleaved images and texts). To be specific, we use monomodal data for representation learning. For example, language modeling with text data pretrains instruction following, in-context learning, and various language tasks. Moreover, cross-modal pairs and interleaved data learn to align the perception of general modalities with language models. Interleaved data also naturally fit in the multimodal language modeling task. We present more details of training data collection in the supplemental material.

The models are trained with the next-token prediction task, i.e., learning to generate the next token depending on the previous context. The training objective is to maximize the log-likelihood of tokens in examples. Notice that only discrete tokens, such as text tokens, are accounted for in the training loss. Multimodal language modeling is a scalable way to train the models. More importantly, the emergence of various capabilities makes the training task favorable for downstream applications.

# 3 Experiments

## 3.1 Training Setup

We train KOSMOS-1 with 1.6 billion parameters using a mix of text corpora, image-caption pairs, and interleaved data. We use Magneto's initialization for optimization stability and a pretrained CLIP ViT-L/14 model for image representation. The model is trained for 300k steps using a batch size of 1.2 million tokens and the AdamW optimizer. We adopt a learning rate warm-up and decay schedule, and use SentencePiece for tokenization.

To improve instruction-following capabilities, we perform language-only instruction tuning using Unnatural Instructions [24] and FLANv2 [25] datasets. This tuning process is conducted as language modeling, and improvements transfer across modalities. More details about hyperparameters can be found in the supplemental material.

Table 1 summarizes the corresponding datasets and what capabilities we would like to evaluate. We evaluate different capabilities related to language, perception-language and vision.

## 3.2 Perception-Language Tasks

**Image Captioning** Table 2a shows the captioning performance on COCO [39] Karpathy test split and Flickr30k [40] test set. KOSMOS-1 achieves remarkable results in zero-shot setting on two image captioning datasets. Specifically, our model achieves a CIDEr score of 67.1 on the Flickr30k dataset, compared to 60.6 and 61.5 for the Flamingo-3B and Flamingo-9B models, respectively. Notably, our model is able to accomplish this feat with a smaller size of 1.6B, compared to Flamingo models. This demonstrates our model's superiority in zero-shot image captioning.

**Visual Question Answering** Table 2b reports the visual question answering results on VQAv2 [41] and VizWiz [42]. We show that KOSMOS-1 can better handle the diversity and complexity of the VizWiz dataset. KOSMOS-1 achieves higher accuracy and robustness than Flamingo-3B and Flamingo-9B models on zero-shot settings. In addition, our model is competitive with Flamingo on the VQAv2 dataset.

| Dataset | Task description | Metric | Zero-shot | Few-shot |
|---|---|---|---|---|
| *Language tasks* | | | | |
| StoryCloze [26] | Commonsense reasoning | Accuracy | ✓ | ✓ |
| HellaSwag [27] | Commonsense NLI | Accuracy | ✓ | ✓ |
| Winograd [28] | Word ambiguity | Accuracy | ✓ | ✓ |
| Winogrande [29] | Word ambiguity | Accuracy | ✓ | ✓ |
| PIQA [30] | Physical commonsense | Accuracy | ✓ | ✓ |
| BoolQ [31] | Question answering | Accuracy | ✓ | ✓ |
| CB [32] | Textual entailment | Accuracy | ✓ | ✓ |
| COPA [33] | Causal reasoning | Accuracy | ✓ | ✓ |
| Rendered SST-2 [34] | OCR-free sentiment classification | Accuracy | ✓ | |
| HatefulMemes [35] | OCR-free meme classification | ROC AUC | ✓ | |
| *Cross-modal transfer* | | | | |
| RelativeSize [36] | Commonsense reasoning (object size) | Accuracy | ✓ | |
| MemoryColor [37] | Commonsense reasoning (object color) | Accuracy | ✓ | |
| ColorTerms [38] | Commonsense reasoning (object color) | Accuracy | ✓ | |
| *Nonverbal reasoning tasks* | | | | |
| IQ Test | Raven's Progressive Matrices | Accuracy | ✓ | |
| *Perception-language tasks* | | | | |
| COCO Caption [39] | Image captioning | CIDEr, etc. | ✓ | ✓ |
| Flicker30k [40] | Image captioning | CIDEr, etc. | ✓ | ✓ |
| VQAv2 [41] | Visual question answering | VQA acc. | ✓ | ✓ |
| VizWiz [42] | Visual question answering | VQA acc. | ✓ | ✓ |
| WebSRC [43] | Web page question answering | F1 score | ✓ | |
| *Vision tasks* | | | | |
| CUB [44] | Zero-shot image classification with descriptions | Accuracy | ✓ | |

Table 1: We evaluate the capabilities of KOSMOS-1 on language, perception-language, and vision tasks under both zero- and few-shot learning settings.

## 3.3 IQ Test: Nonverbal Reasoning

Raven's Progressive Matrices [9, 8] is one of the most common tests to evaluate nonverbal reasoning. The capability of nonverbal reasoning is typically a reflection of an individual's intelligence quotient (IQ). Figure 2 shows an example. Given eight images, the task is to identify the following element from six similar candidates.

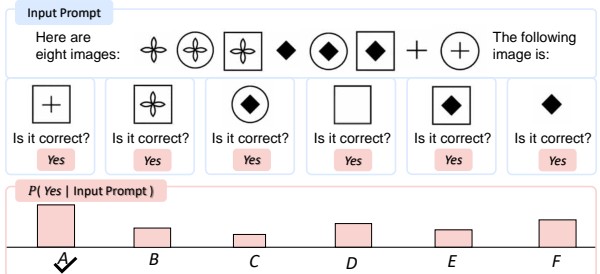

| Method | Accuracy |
|---|---|
| Random Choice | 17% |
| KOSMOS-1 | **22%** |

Figure 2: We append each candidate image to the prompt separately and query the model if it is correct.

Table 3: Zero-shot generalization on Raven IQ test.

The models need to conduct zero-shot nonverbal reasoning without explicitly fine-tuning. The Raven IQ test is analogous to in-context learning of language models, where the difference is whether the context is nonverbal or verbal. In order to infer the answers, the models have to recognize abstract concepts and identify the underlying patterns of given images. So the IQ task is a good testbed to benchmark the nonverbal in-context learning capability.

Table 3 shows the evaluation results on the IQ test dataset. KOSMOS-1 achieves 5.3% improvement respectively over the random baseline. The results indicate that KOSMOS-1 is able to perceive abstract conceptual patterns in a nonverbal context, and then deduce the following element across multiple choices. To the best of our knowledge, it is the first time that a model can perform such zero-shot Raven IQ tests. Although there is still a large performance gap between the current model and the

| Shot | Model | COCO | Flickr30k |
|---|---|---|---|
| 0 | ZeroCap [45] | 14.6 | - |
| | VLKD [46] | 58.3 | - |
| | FewVLM [47] | - | 31.0 |
| | METALM [3] | 82.2 | 43.4 |
| | Flamingo-3B* [5] | 73.0 | 60.6 |
| | Flamingo-9B* [5] | 79.4 | 61.5 |
| | KOSMOS-1 (1.6B) | **84.7** | **67.1** |
| 2 | Flamingo-3B* [5] | - | - |
| | Flamingo-9B* [5] | - | - |
| | KOSMOS-1 (1.6B) | **99.6** | **70.0** |
| 4 | Flamingo-3B* [5] | 85.0 | 72.0 |
| | Flamingo-9B* [5] | 93.1 | 72.6 |
| | KOSMOS-1 (1.6B) | **101.7** | **75.3** |
| 8 | Flamingo-3B* [5] | 90.6 | 71.7 |
| | Flamingo-9B* [5] | **99.0** | **73.4** |
| | KOSMOS-1 (1.6B) | 96.7 | 68.0 |

| Shot | Model | VQAv2 | VizWiz |
|---|---|---|---|
| 0 | Frozen | 29.5 | - |
| | VLKDViT-B/16 | 38.6 | - |
| | METALM | 41.1 | - |
| | Flamingo-3B* | 49.2 | 28.9 |
| | Flamingo-9B* | **51.8** | 28.8 |
| | KOSMOS-1 (1.6B) | 51.0 | **29.2** |
| 2 | Flamingo-3B* [5] | - | - |
| | Flamingo-9B* [5] | - | - |
| | KOSMOS-1 (1.6B) | **51.4** | **31.4** |
| 4 | Flamingo-3B* [5] | 53.2 | 34.4 |
| | Flamingo-9B* [5] | **56.3** | 34.9 |
| | KOSMOS-1 (1.6B) | 51.8 | **35.3** |
| 8 | Flamingo-3B* [5] | 55.4 | 38.4 |
| | Flamingo-9B* [5] | **58.0** | **39.4** |
| | KOSMOS-1 (1.6B) | 51.4 | 39.0 |

(a) Image captioning results on COCO caption Karpathy test and Flickr30k test. We present CIDEr scores.

(b) Visual question answering results on VQAv2 and VizWiz. We present VQA accuracy scores.

Table 2: "*": Flamingo [5] builds the zero-shot prompt with two examples from the downstream tasks where their corresponding images are removed (i.e., similar to few-shot text prompts) while the others evaluate true zero-shot learning.

average level of adults, KOSMOS-1 demonstrates the potential of MLLMs to perform zero-shot nonverbal reasoning by aligning perception with language models.

## 3.4 OCR-Free Language Understanding

OCR-free language understanding is a task that focuses on understanding text and images without relying on Optical Character Recognition (OCR). For example, during the Rendered SST-2 task [34], sentences from the Stanford Sentiment Treebank [48] dataset are rendered as images. The model is asked to predict the sentiment of the text within the images. The task evaluates a model's ability to read and comprehend the meaning of words and sentences directly from the images.

As shown in Table 4a, KOSMOS-1 achieves a ROC AUC of 63.9% for the HatefulMemes validation set and a test accuracy of 67.1% for the Rendered SST-2 test set. It outperforms CLIP ViT-L and Flamingo-9B, which achieve AUCs of 63.3% and 57.0% on the HatefulMemes task. Note that Flamingo explicitly provides OCR text into the prompt, while KOSMOS-1 does not access any external tools or resources. This indicates that KOSMOS-1 has built-in abilities to read and comprehend the text in the rendered images.

| Model | HatefulMemes | Rendered SST-2 |
|---|---|---|
| CLIP ViT-B/32 | 57.6 | 59.6 |
| CLIP ViT-B/16 | 61.7 | 59.8 |
| CLIP ViT-L/14 | 63.3 | 64.0 |
| Flamingo-3B | 53.7 | - |
| Flamingo-9B | 57.0 | - |
| KOSMOS-1 (1.6B) | **63.9** | **67.1** |

| Model | EM | F1 |
|---|---|---|
| *Using extracted text* | | |
| LLM | 7.6 | 17.9 |
| KOSMOS-1 | **15.8** | **31.3** |
| *Without using extracted text* | | |
| KOSMOS-1 | 3.8 | 10.6 |

(a) Zero-shot generalization on OCR-free language understanding. We report accuracy scores.

(b) Zero-shot performance on WebSRC task. We report exact match (EM) and F1 scores.

## 3.5 Web Page Question Answering

Web page question answering aims at finding answers to questions from web pages. It requires the model to comprehend both the semantics and the structure of texts (such as tables, lists, and

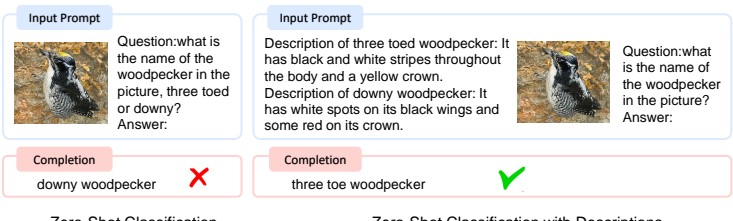

Figure 3: In-context verbal descriptions can help KOSMOS-1 recognize visual categories better.

HTML layout). We compare the performance on the Web-based Structural Reading Comprehension (WebSRC) dataset [43]. For comparisons, we train a language model (LLM) on the same text corpora with the same training setup as in KOSMOS-1.

The experimental results are summarized in Table 4b. We observe that KOSMOS-1 outperforms the LLM, indicating that KOSMOS-1 can benefit from the layout and style information of web pages in images. In addition, we evaluate the performance of KOSMOS-1 without the extracted text in the prompt. It shows that extracted text has a contribution of +12.0/20.7 EM/F1 to KOSMOS-1, indicating that the benefit from modeling images does not sacrifice its language abilities.

## 3.6  Multimodal Chain-of-Thought Prompting

Chain-of-thought prompting [10] allows large language models to generate a series of reasoning steps and decompose a multi-step problem into intermediate steps, which can significantly improve the performance in complex tasks. Motivated by chain-of-thought prompting, we investigate a multimodal chain-of-thought prompting using KOSMOS-1. We break down perception-language tasks into two steps. In the first stage, given an image, we use a prompt to guide the model to generate a rationale. The model is then fed the rationale and a task-aware prompt to produce the final results.

We conduct experiments to evaluate the performance of the multimodal chain-of-thought prompting. Table 5a shows that multimodal chain-of-thought prompting achieves a score of 72.9, which is 5.8 points higher than the standard prompting. By generating intermediate content, the model can recognize the text in the images and infer the sentiment of the sentences more correctly.

| Model | Accuracy |
|---|---|
| CLIP ViT-B/32 | 59.6 |
| CLIP ViT-B/16 | 59.8 |
| CLIP ViT-L/14 | 64.0 |
| KOSMOS-1 | 67.1 |
|    w/ multimodal CoT prompting | **72.9** |

(a) Multimodal chain-of-thought (CoT) prompting on Rendered SST-2 task.

| Settings | Accuracy |
|---|---|
| Without Descriptions | 61.7 |
| With Descriptions | **90.0** |

(b) Results of zero-shot image classification without and with verbal descriptions.

## 3.7  Zero-Shot Image Classification with Descriptions

The standard approach of image classification as above is to prompt the model for the specific name of the object depicted in the image. However, there are also some classification rules customized for different users and scenarios, such as the refined classification of complex animal subspecies. We can utilize natural language descriptions to guide KOSMOS-1 to distinguish images in the zero-shot setting, which makes the decision process more interpretable. Following CUB [44], we construct a bird classification dataset that contains images and natural-language descriptions of categories. The evaluation procedure is illustrated in Figure 3.

The evaluation results are shown in Table 5b. We observe that providing descriptions in context can significantly improve the accuracy of image classification. The consistent improvements indicate that KOSMOS-1 can perceive the intentions of instructions and well align the concepts in language modality with visual features in vision modality.

## 3.8 Language Tasks

The models are evaluated on the language tasks given task instructions (i.e., zero-shot) or several demonstration examples (i.e., few-shot). Text inputs are directly fed into the models as in vanilla language models. We train a language model (LLM) baseline with the same text corpora and training setup. We evaluate KOSMOS-1 and the LLM baseline on eight language tasks.

Table 6 presents the in-context learning performance of language tasks. KOSMOS-1 achieves comparable or even better performance in cloze completion and commonsense reasoning tasks when compared to LLM. In terms of the average result across all these datasets, LLM performs better in zero-shot and one-shot settings, whereas our model performs better in few-shot ($k = 4$) settings. In addition, Section 3.9.2 shows that MLLMs learn better visual commonsense knowledge compared with LLMs.

| Task | Zero-shot | | One-shot | | Few-shot ($k = 4$) | |
|------|-----------|-----------|----------|-----------|--------|-----------|
| | LLM | KOSMOS-1 | LLM | KOSMOS-1 | LLM | KOSMOS-1 |
| StoryCloze | **72.9** | 72.1 | **72.9** | 72.2 | **73.1** | 72.3 |
| HellaSwag | **50.4** | 50.0 | **50.2** | 50.0 | **50.4** | 50.3 |
| Winograd | **71.6** | 69.8 | **71.2** | 68.4 | **70.9** | 69.8 |
| Winogrande | **56.7** | 54.8 | **56.7** | 54.5 | **57.0** | 55.7 |
| PIQA | **73.2** | 72.9 | **73.0** | 72.5 | **72.6** | 72.3 |
| BoolQ | **56.4** | **56.4** | 55.1 | **57.2** | 58.7 | **59.2** |
| CB | 39.3 | **44.6** | 41.1 | **48.2** | 42.9 | **53.6** |
| COPA | **68.0** | 63.0 | **69.0** | 64.0 | **69.0** | 64.0 |
| Average | 61.1 | 60.5 | 61.2 | 60.9 | 61.8 | 62.2 |

Table 6: Performance comparisons of language tasks between KOSMOS-1 and LLM. We use the same textual data and training setup to reimplement a language model. Both models do not use instruction tuning for fair comparisons.

## 3.9 Cross-modal Transfer

Cross-modal transferability allows a model to learn from one modality (such as text, image, audio, etc.) and transfer the knowledge to the other modalities. This skill can enable a model to perform various tasks across different modalities. In this part, we evaluate the cross-model transferability of KOSMOS-1 on several benchmarks.

### 3.9.1 Transfer from Language to Multimodal: Language-Only Instruction Tuning

To evaluate the effect of language-only instruction tuning, we conduct an ablation study using four datasets: COCO, Flickr30k, VQAv2, and VizWiz. These datasets consist of image captioning and visual questions answsering. The evaluation metrics are: CIDEr scores for COCO/Flickr30k and VQA accuracy for VQAv2/VizWiz.

Table 7 shows the experimental results. Language-only instruction tuning boosts our model's performance by 1.9 points on Flickr30k, 4.3 points on VQAv2, and 1.3 points on VizWiz. Our experiments show that language-only instruction tuning can significantly improve the model's instruction-following capabilities across modalities. The results also indicate that our model can transfer the instruction-following capability from language to other modalities.

| Model | COCO | Flickr30k | VQAv2 | VizWiz |
|-------|------|-----------|-------|--------|
| KOSMOS-1 | 84.7 | **67.1** | **51.0** | **29.2** |
| w/o language-only instruction tuning | **87.6** | 65.2 | 46.7 | 27.9 |

Table 7: Ablation study on language-only instruction tuning. We report CIDEr scores for COCO and Flickr30k, and VQA accuracy scores for VQAv2 and VizWiz.

### 3.9.2 Transfer from Multimodal to Language: Visual Commonsense Reasoning

Visual commonsense reasoning tasks require an understanding of the properties of everyday objects in the real world, such as color, size, and shape. These tasks are challenging for language models because they may require more information about object properties than what is available in texts. To investigate the visual commonsense capabilities, we compare the zero-shot performance of KOSMOS-1 and LLM on three object commonsense reasoning datasets, RELATIVESIZE [36], MEMORYCOLOR [37] and COLORTERMS [38] datasets. RELATIVESIZE contains 486 object pairs from 41 physical objects. The model is required to predict the size relation between two objects in a binary question-answering format with "Yes"/"No" answers. MEMORYCOLOR and COLORTERMS require the model to predict the color of objects from a set of 11 color labels in a multiple-choice format.

Table 8 presents the zero-shot performance of KOSMOS-1 and LLM on visual commonsense reasoning tasks. KOSMOS-1 significantly outperforms LLM by 1.5% on RELATIVESIZE, 14.7% on MEMORYCOLOR, and 9.7% on COLORTERMS dataset. The consistent improvements indicate that KOSMOS-1 benefits from the visual knowledge to complete the corresponding visual commonsense reasoning. The reason for KOSMOS-1's superior performance is that it has modality transferability, which enables the model to transfer visual knowledge to language tasks. On the contrary, LLM has to rely on textual knowledge and clues to answer visual commonsense questions, which limits its ability to reason about object properties.

| Model | Size Reasoning | Color Reasoning | |
| | RELATIVESIZE | MEMORYCOLOR | COLORTERMS |
|---|---|---|---|
| *Using retrieved images* | | | |
| VALM [49] | 85.0 | 58.6 | 52.7 |
| *Language-only zero-shot evaluation* | | | |
| LLM | 92.7 | 61.4 | 63.4 |
| KOSMOS-1 | **94.2** | **76.1** | **73.1** |

Table 8: Zero-shot visual commonsense reasoning on RELATIVESIZE, MEMORYCOLOR, and COLORTERMS datasets. Accuracy scores are reported.

## 4 Related Work

In recent years, vision-language learning and representation models has garnered significant attention [2, 3, 4, 34, 50, 51, 52, 53, 54]. Previous vision-language models still exhibit limitations in instruction following, in-context abilities, and generalization capabilities for unseen tasks. Researchers have begun exploring more powerful multimodal large language models. Flamingo [5] trained its model from scratch and made it possible to generate text tokens conditioned on both visual and text inputs. Another category of research focus on learning multimodality abilities based on LLMs [7, 55, 56]. Meanwhile, some work [57, 58, 59] introduce visual instruction tuning to enhance instruction following capabilities.

## 5 Conclusion

In this work, we introduce KOSMOS-1, a multimodal large language model that can perceive general modalities, follow instructions, and perform in-context learning. The models trained on web-scale multimodal corpora achieve promising results across a wide range of language tasks and multimodal tasks. We show that going from LLMs to MLLMs enables new capabilities and opportunities. In the future, we would like to scale up KOSMOS-1 in terms of model size [13, 14, 60], and integrate the speech [12] capability into KOSMOS-1. In addition, KOSMOS-1 can be used as a unified interface for multimodal learning, e.g., enabling using instructions and examples to control text-to-image generation. We further discuss the limitations and broader societal impacts of KOSMOS-1 in the supplemental material.

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
