# A  Limitations and Societal Impacts

**Limitations**   One limitation of our model is its potential for data bias. KOSMOS-1 is trained on a web-scale multimodal corpus, which means that it is likely to be biased towards the data that it was trained on. This could lead to the model generating text that is biased towards certain demographics or viewpoints.

Another limitation of KOSMOS-1 is its relatively small size compared to other large language models. This means that the model may not be able to learn as complex relationships between different modalities. This could lead to the model making mistakes when it is asked to perform tasks that require a deep understanding of multiple modalities.

Finally, KOSMOS-1 only supports vision modality. This means that the model cannot process other modalities such as speech. This could limit the applications of the model.

**Societal Impacts**   The broader impact of this paper is that it introduces a new type of large language model that can perceive general modalities, follow instructions, and perform in-context learning. This has the potential to be used for a variety of beneficial applications, such as new educational tools and interactive dialogue assistants in video games. However, there are also potential negative impacts of MLLMs. MLLMs could be used to create fake news articles or social media posts. MLLMs could be used to generate text that reveals private information from web-scale pre-training data.

# B  Hyperparameters

## B.1  Training

We report the detailed model hyperparameter settings of KOSMOS-1 in Table 1 and training hyperparameters in Table 2.

| Hyperparameters | |
|---|---|
| Number of layers | 24 |
| Hidden size | 2,048 |
| FFN inner hidden size | 8,192 |
| Attention heads | 32 |
| Dropout | 0.1 |
| Attention dropout | 0.1 |
| Activation function | GeLU [1] |
| Vocabulary size | 64,007 |
| Soft tokens $V$ size | 64 |
| Max length | 2,048 |
| Relative position embedding | xPos [2] |
| Initialization | Magneto [3] |

Table 1: Hyperparameters of causal language model of KOSMOS-1

## B.2  Language-Only Instruction Tuning

The detailed instruction tuning hyperparameters are listed in Table 3.

# C  Datasets

## C.1  Pretraining

The models are trained on web-scale multimodal corpora. The training datasets consist of text corpora, image-caption pairs, and interleaved data of images and texts.

| Hyperparameters | |
| --- | --- |
| Training steps | 300,000 |
| Warmup steps | 375 |
| Batch size of text corpora | 256 |
| Max length of text corpora | 2,048 |
| Batch size of image-caption pairs | 6,144 |
| Batch size of interleaved data | 128 |
| Optimizer | Adam |
| Learning rate | 2e-4 |
| Learning Rate Decay | Linear |
| Adam $\epsilon$ | 1e-6 |
| Adam $\beta$ | (0.9, 0.98) |
| Weight decay | 0.01 |

Table 2: Training hyperparameters of KOSMOS-1

| Hyperparameters | |
| --- | --- |
| Training steps | 10,000 |
| Warmup steps | 375 |
| Batch size of instruction data | 256 |
| Batch size of text corpora | 32 |
| Batch size of image-caption pairs | 768 |
| Batch size of interleaved data | 16 |
| Learning rate | 2e-5 |

Table 3: Instruction tuning hyperparameters of KOSMOS-1

**Text Corpora**   We train our model with The Pile [4] and Common Crawl (CC). The Pile is a massive English text dataset built for training large-scale language models, which is produced from a variety of data sources. We exclude data splits from GitHub, arXiv, Stack Exchange, and PubMed Central. We also include the Common Crawl snapshots (2020-50 and 2021-04) datasets, CC-Stories, and RealNews datasets [5, 6]. The entire datasets have been purged of duplicate and near-duplicate documents, as well as filtered to exclude downstream task data.

Table 4 provides a full overview of the language datasets that were used in the training of KOSMOS-1 model. These data sources can be divided into the following three categories:

- **Academic**: NIH Exporter
- **Internet**: Pile-CC, OpenWebText2, Wikipedia (English), CC-2020-50, CC-2021-04, Realnews
- **Prose**: BookCorpus2, Books3, Gutenberg [7], CC-Stories

**Image-Caption Pairs**   The image-caption pairs are constructed from several datasets, including English LAION-2B [8], LAION-400M [9], COYO-700M [10], and Conceptual Captions [11, 12]. English LAION-2B, LAION-400M, and COYO-700M are collected from web pages of the Common Crawl web data by extracting image sources and the corresponding alt-text. Conceptual Captions are also from internet web pages.

LAION-2B contains about 2B English image-caption pairs, LAION-400M consists of 400M English image-caption pairs, and COYO-700M has 700M English image-caption pairs. Conceptual Captions contains 15M English image-caption pairs and consists of two datasets: CC3M and CC12M, which are also collected from internet webpages using a Flume pipeline. For Conceptual Captions, we discard pairs whose captions contain special tags such as "<PERSON>".

**Interleaved Image-Text Data**   We collect interleaved multimodal data from the Common Crawl snapshot, which is a publicly available archive of web pages. We use a filtering process to select about 71M web pages from the original 2B web pages in the snapshot. We then extract the text and

| Datasets | Tokens (billion) | Weight (%) | Epochs |
|---|---|---|---|
| OpenWebText2 | 14.8 | 21.8% | 1.47 |
| CC-2021-04 | 82.6 | 17.7% | 0.21 |
| Books3 | 25.7 | 16.2% | 0.63 |
| CC-2020-50 | 68.7 | 14.7% | 0.21 |
| Pile-CC | 49.8 | 10.6% | 0.21 |
| Realnews | 21.9 | 10.2% | 0.46 |
| Wikipedia | 4.2 | 5.4% | 1.29 |
| BookCorpus2 | 1.5 | 1.1% | 0.75 |
| Gutenberg (PG-19) | 2.7 | 1.0% | 0.38 |
| CC-Stories | 5.3 | 1.0% | 0.19 |
| NIH ExPorter | 0.3 | 0.2% | 0.75 |

Table 4: Language datasets used to train the KOSMOS-1 model.

images from the HTML of each selected web page. For each document, we limit the number of images to five to reduce noise and redundancy. We also randomly discard half of the documents that only have one image to increase the diversity. By using this corpus, we enable KOSMOS-1 to handle interleaved text and image and improve its few-shot ability.

To ensure quality and relevance, we apply several filtering criteria. First, we discard any pages that are not written in English. Second, we discard any pages that do not have images interspersed in the text. Third, we discard any images that have a resolution lower than 64 by 64 pixels or that are single-colored. Fourth, we discard any text that is not meaningful or coherent, such as spam or gibberish. We use some heuristics to identify and remove gibberish text containing emoji symbols, hashtags, and URL links. After applying these filters, we end up with about 71 million documents for training.

## C.2 Data Format

The training data is organized in the format as follows:

| Datasets | Format Examples |
|---|---|
| **Text** | `` KOSMOS-1 can perceive multimodal input, learn in context, and generate output. `` |
| **Image-Caption** | ` <image>` Image Embedding `</image>` WALL-E giving potted plant to EVE. `` |
| **Multimodal** | ` <image>` Image Embedding `</image>` This is WALL-E. `<image>` Image Embedding `</image>` This is EVE. `` |

Table 5: The examples of the data format to train the KOSMOS-1 model.

# D  Evaluation

## D.1  Input Format Used for Perception-Language Tasks

Figure 1 shows how we conduct zero-shot and few-shot evaluations on perception-language tasks.

## D.2  Perception-Language Tasks

We evaluate the caption generation on MS COCO Caption [13], and Flickr30k [14]. We use the test set of COCO *Karpathy split* [15], which re-partitions the train2014 and val2014 images [13] into 113,287, 5,000, and 5,000 for the training set, validation set, and test set, respectively. We conduct an evaluation on Flickr30k's *Karpathy split* test set. The image resolution is 224×224. We use beam search to generate the captions, and the beam size is 5. In the few-shot settings, we randomly

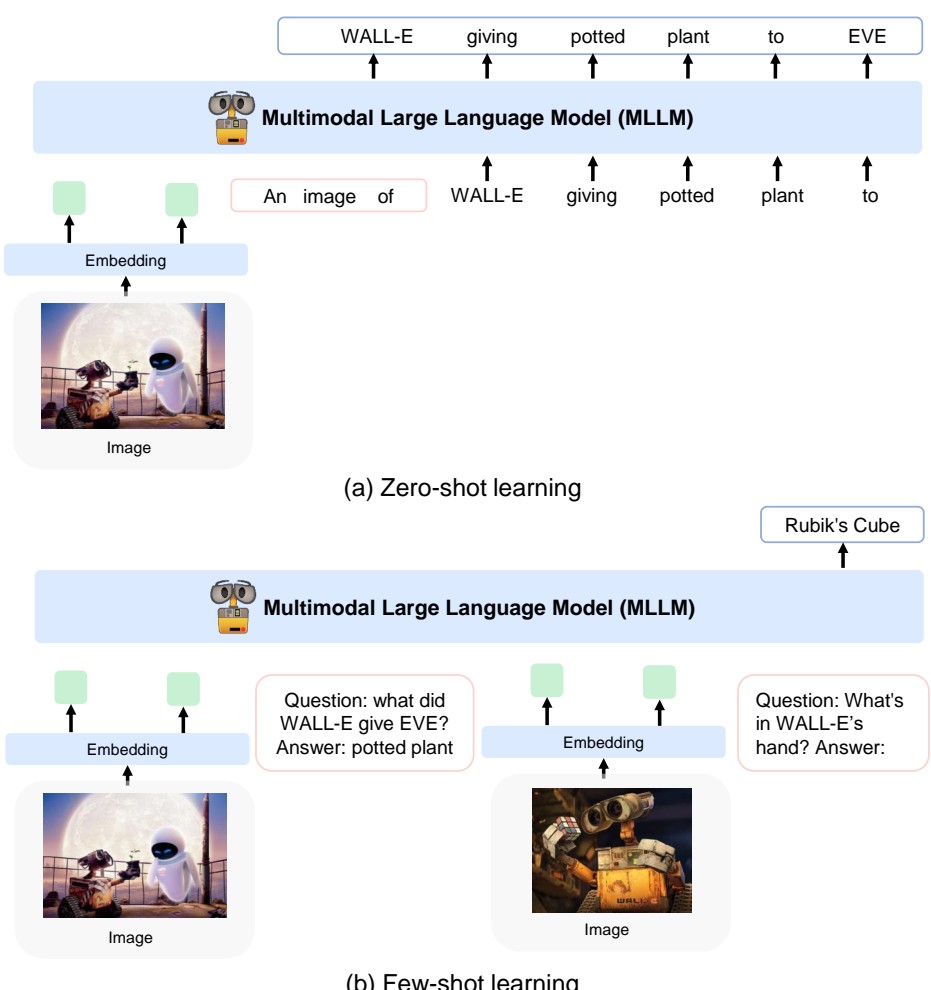

Figure 1: We evaluate KOSMOS-1 on the perception-language tasks in zero- and few-shot settings. (a) Zero-shot learning, e.g., zero-shot image captioning with language prompts. (b) Few-shot learning, e.g., visual question answering with in-context learning.

sample demonstrations from the training set. We use COCOEvalCap[1] to compute CIDEr [16] and SPICE [17] scores as the evaluation metrics. We prompt KOSMOS-1 with *"An image of"* for zero-shot and few-shot caption generation experiments.

For visual question-answering tasks, we evaluate zero-shot and few-shot results on test-dev set of VQAv2 [18] and test-dev set of VizWiz [19], respectively. The resolution of images is 224×224. We use greedy search for the decoding. We follow the normalization rules of the VQAv2 evaluation code[2] when computing the VQA accuracy. We evaluate the performance of VQA in an open-ended setting that KOSMOS-1 generates answers and stops at the `` ("end of sequence") token. The prompt is *"Question: {question} Answer: {answer}"* for visual question answering tasks.

---

[1] https://github.com/salaniz/pycocoevalcap
[2] https://github.com/GT-Vision-Lab/VQA

## D.3    IQ Test Tasks

To evaluate the KOSMOS-1 on zero-shot nonverbal reasoning, we construct a dataset of the Raven IQ test. It consists of 50 examples collected from different websites[3456]. Each example has three (i.e., $2 \times 2$ matrix), four, or eight (i.e., $3 \times 3$ matrix) given images. The goal is to predict the next one. Each instance has six candidate images with a unique correct completion. We measure accuracy scores to evaluate the models. The evaluation dataset is available at https://aka.ms/kosmos-iq50.

The matrix-style images are flattened and fed into the models one-by-one. To enable the model to better understand the desired task, we also use a textual instruction *"Here are three/four/eight images:"*, *"The following image is:"*, and *"Is it correct?"* for conditioning. We append each possible candidate to the context separately and compare the probability that the model outputs "Yes" in a close-ended setting. The candidate that yields the largest probability is regarded as the prediction.

## D.4    OCR-Free Tasks

We evaluate OCR-free language understanding on the Rendered SST-2 [20] test set and Hateful-Memes [21] validation set. We use accuracy as the metric for the Rendered SST-2 and report ROC AUC for the HatefulMemes dataset. We use the prompt *"Question: what is the sentiment of the opinion? Answer: {answer}"*, where the answer is either positive or negative for the Rendered SST-2. For the HatefulMemes task, the prompt is *"Question: does this picture contain real hate speech? Answer: {answer}"*, where the answer is either yes or no.

## D.5    Web Page Tasks

We compare the performance on the Web-based Structural Reading Comprehension (WebSRC) dataset [22]. For comparisons, we train a language model (LLM) on the same text corpora with the same training setup as in KOSMOS-1. The LLM takes the text extracted from the web page as input. Its template of the prompt is *"Given the context below from web page, extract the answer from the given text like this: Qusestion: Who is the publisher of this book? Answer: Penguin Books Ltd. Context: {WebText} Q: {question} A: {answer} "*, where the *{WebText}* presents the text extracted from the web page. Besides using the same prompt, KOSMOS-1 prepends the image before the prompt. Two example images from WebSRC are shown in Appendix D.11. Following the original paper [22], we use exact match (EM) and F1 scores as our evaluation metrics.

## D.6    Multimodal CoT Tasks

We evaluate the ability of multimodal chain-of-thought prompting on the Rendered SST-2. We use the prompt *"Introduce this picture in detail:"* to generate the content in the picture as the rationale. Then, we use the prompt *"{rationale} Question: what is the sentiment of the opinion? Answer: {answer}"* to predict the sentiment, where the answer is either positive or negative.

## D.7    Zero-shot image classification Tasks

Given an input image, we concatenate the image with the prompt *"The photo of the"*. The input is then fed into the model to obtain the category name of the image. We evaluate the model on ImageNet [23], which contains 1.28M training images and 50k validation images in 1k object categories. The prediction is classified as correct if it is exactly the same as the ground-truth category name. The image resolution used for evaluation is 224×224. We use beam search to generate the category names and the beam size is 2.

## D.8    Zero-Shot Image Classification with Descriptions

Following CUB [24], we construct a bird classification dataset that contains images and natural-language descriptions of categories. The dataset has three groups of binary image classification. Each

---

[3] https://en.testometrika.com/intellectual/iq-test/
[4] https://en.testometrika.com/intellectual/iq-test-for-kids-7-to-16-year-old/
[5] https://iqpro.org/
[6] https://iqhaven.com/matrix-g

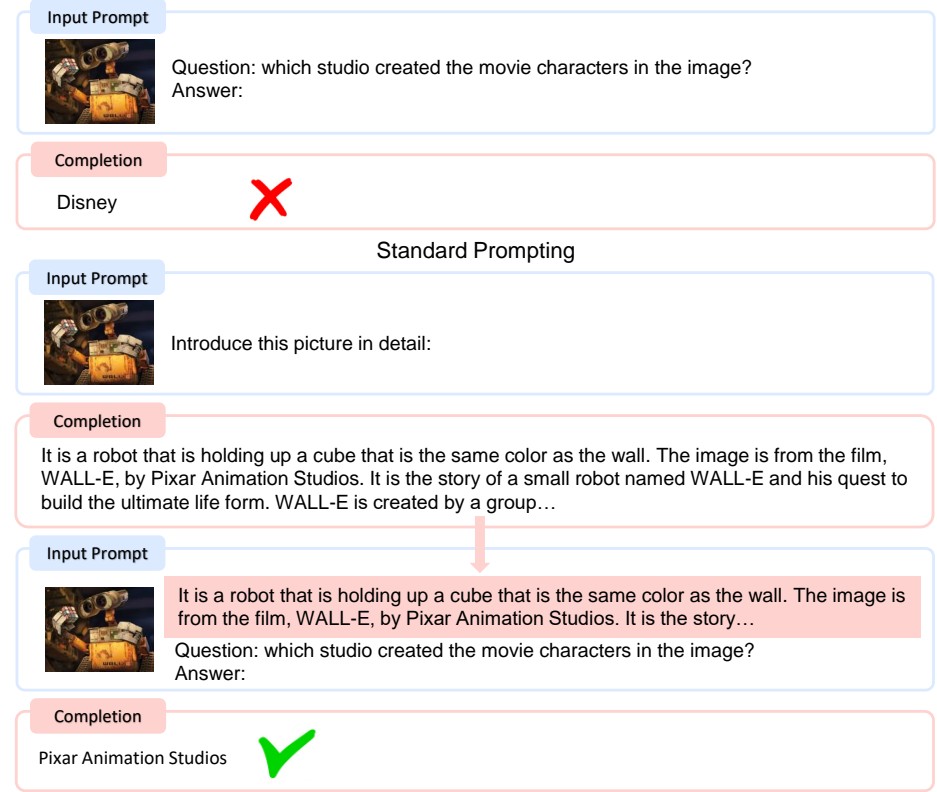

Standard Prompting

Multimodal Chain-of-Thought Prompting

Figure 2: Multimodal Chain-of-Thought prompting enables KOSMOS-1 to generate a rationale first, then to tackle complex question-answering and reasoning tasks.

group contains two animal categories with similar appearances. Our goal is to classify images given the categories' descriptions. Table 6 presents the data samples. The first group is from [24], while the other two groups are collected from the website. Each category contains twenty images.

The evaluation procedure is illustrated in Figure **??**. For the zero-shot setting, we provide detailed descriptions of two specific categories and use the template *"Question:what is the name of {general category} in the picture? Answer:"* to prompt the model for the specific category name in an open-ended manner. To evaluate the effect of providing verbal descriptions in context, we also implement a zero-shot baseline without prompting descriptions. Instead, we provide the corresponding specific names in the prompt.

## D.9   Cross-modal Transfer task

We compare KOSMOS-1 and the LLM baseline on three object commonsense reasoning datasets, RELATIVESIZE [25], MEMORYCOLOR [26] and COLORTERMS [27] datasets. Table 7 shows some examples of object size and color reasoning tasks. RELATIVESIZE contains 486 object pairs from 41 physical objects. The model is required to predict the size relation between two objects in a binary question-answering format with "Yes"/"No" answers. MEMORYCOLOR and COLORTERMS require the model to predict the color of objects from a set of 11 color labels in a multiple-choice format. We use only text as our input and do not include any images. We measure the accuracy of our model on these three datasets.

## D.10   Language Tasks

We train a language model (LLM) baseline with the same text corpora and training setup. We evaluate KOSMOS-1 and the LLM baseline on eight language tasks, including cloze and completion tasks (i.e,

| Category 1 | | Category 2 | |
|---|---|---|---|
| three toed woodpecker | | downy woodpecker | |
| 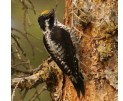 | It has black and white stripes throughout the body and a yellow crown. | 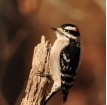 | It has white spots on its black wings and some red on its crown. |
| Gentoo penguin | | royal penguin | |
| 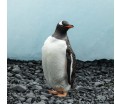 | It has a black head and white patch above its eyes. | 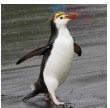 | It has a white face and a yellow crown. |
| black throated sparrow | | fox sparrow | |
| 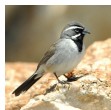 | It has white underparts and a distinctive black bib on the throat. | 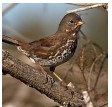 | It has a reddish-brown plumage and a streaked breast. |

Table 6: The detailed descriptions of different categories for in-context image classification.

| Task | Example Prompt | Object / Pair | Answer |
|---|---|---|---|
| Object Size Reasoning | *Is {Item1} larger than {Item2}? {Answer}* | (*sofa*, *cat*) | *Yes* |
| Object Color Reasoning | *The color of {Object} is? {Answer}* | *the sky* | *blue* |

Table 7: Evaluation examples of object size and color reasoning.

StoryCloze, HellaSwag), Winograd-style tasks (i.e, Winograd, Winogrande), commonsense reasoning (i.e, PIQA), and three datasets BoolQ, CB, and COPA from the SuperGLUE benchmark [28]. The detailed descriptions of these datasets are provided in Appendix D.10. We conduct experiments under zero-shot and few-shot settings. We evaluate each test example by randomly sampling examples from the training set as demonstrations. We set the number of shots to 0, 1, and 4 in our experiments.

We conduct experiments on language tasks in four categories:

- Cloze and completion tasks: StoryCloze [29], HellaSwag [30]
- Winograd-style tasks: Winograd [31], Winogrande [32]
- Commonsense reasoning: PIQA [33]
- Three datasets from SuperGLUE benchmark [28]: BoolQ [34], CB [35], COPA [36]

## D.11 WebSRC Task Examples

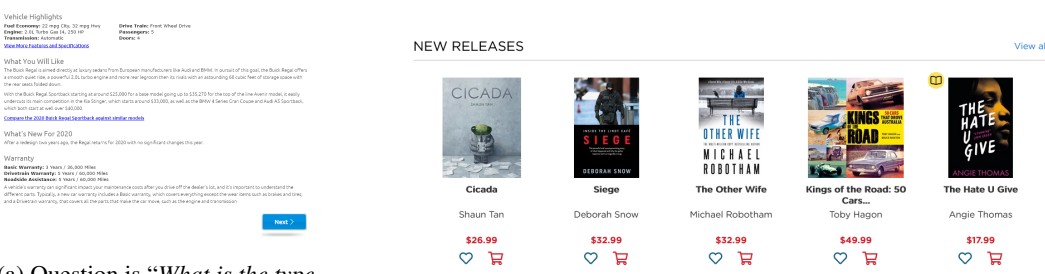

(a) Question is "*What is the type of this drive?*"

(b) Question is "*Who is the author of "Cicada"?*"

Figure 3: Examples form WebSRC [22].

 **D.12   Selected Examples**

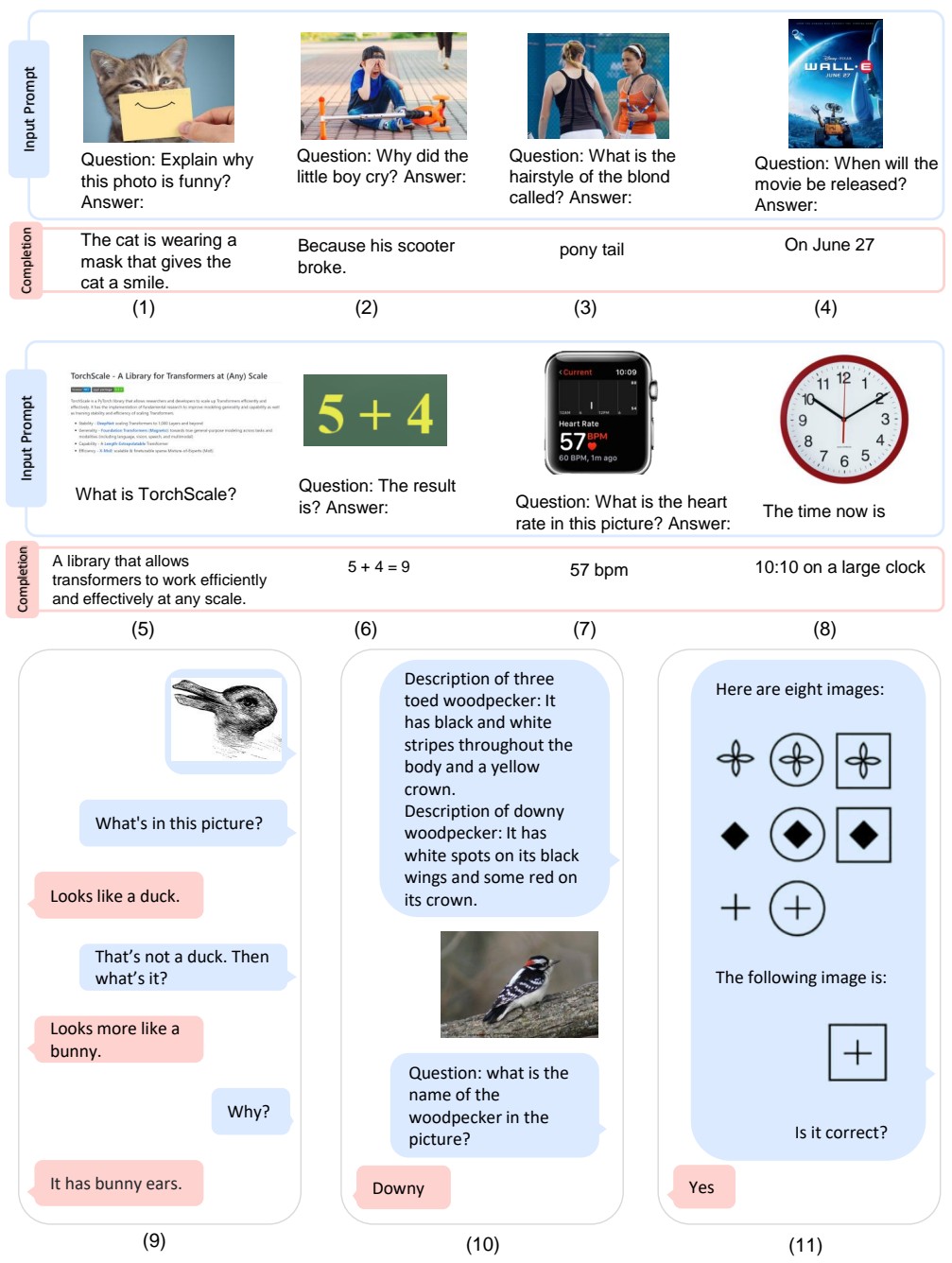

Figure 4: Selected examples generated from KOSMOS-1. Blue boxes are input prompt and pink boxes are KOSMOS-1 output. The examples include (1)-(2) visual explanation, (3)-(4) visual question answering, (5) web page question answering, (6) simple math equation, and (7)-(8) number recognition, and (9)-(11) visual dialogue.

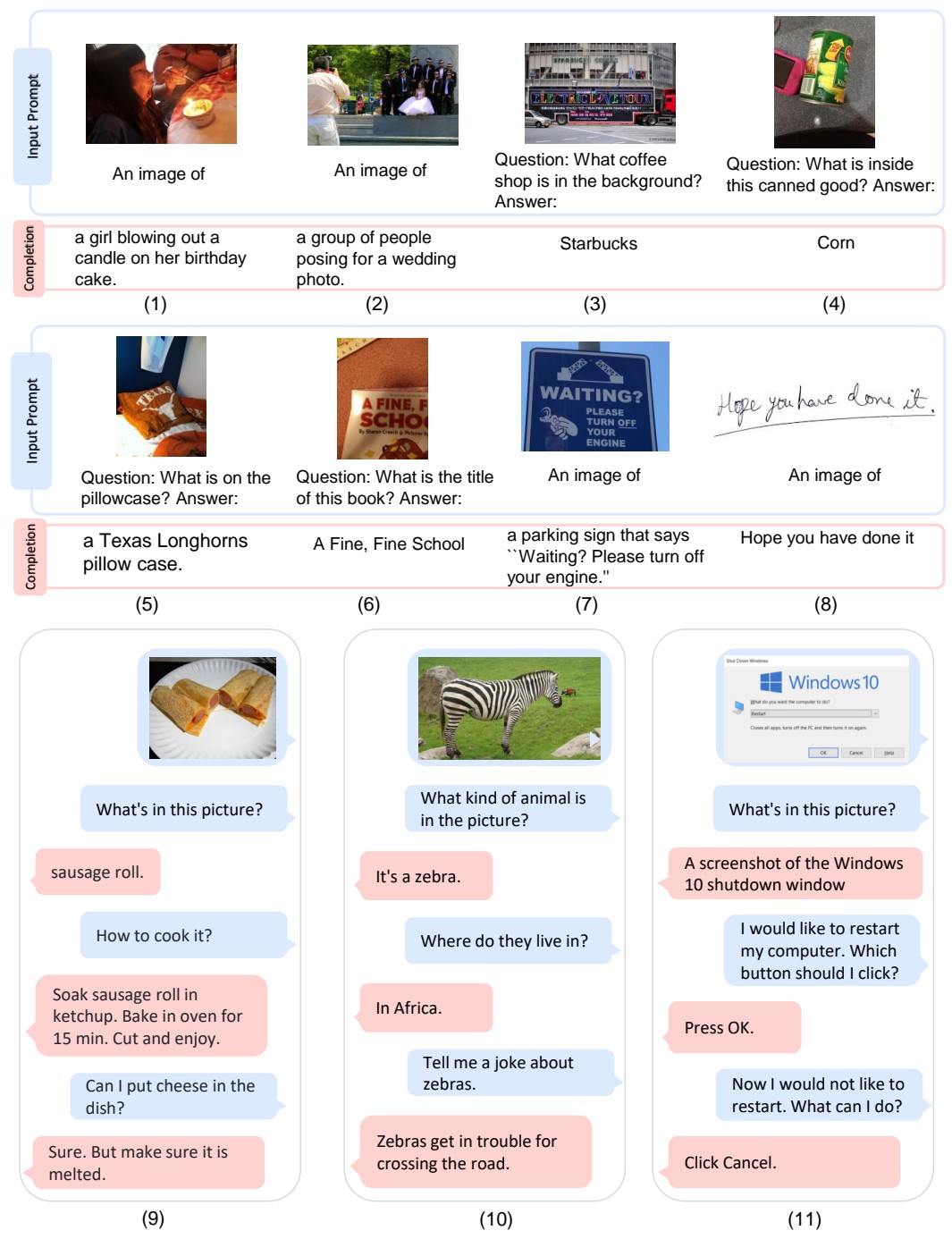

Figure 5: Selected examples generated from KOSMOS-1. Blue boxes are input prompt and pink boxes are KOSMOS-1 output. The examples include (1)-(2) image captioning, (3)-(6) visual question answering, (7)-(8) OCR, and (9)-(11) visual dialogue.