# OpenReview forum: "Language Is Not All You Need: Aligning Perception with Language Models"
_NeurIPS.cc/2023/Conference — NeurIPS 2023 poster_

### Official Review · Reviewer_BRko · 2023-06-12

**Soundness:** 3 good
**Presentation:** 3 good
**Contribution:** 2 fair
**Rating:** 6
**Confidence:** 4

**Summary:**

This paper propose a pretrained multimodal large language model, which can achieve impressive zero-shot performance on many downstream tasks.

The experimental results verify the transfer knowledge from language to multimodal or from multimodal to language.

However, compared to the very similar model FROMAGe and BLIP-2, this model structure seems to be of little difference. The most advantage of this model is that it uses more multimodal data.

It seems that the performance on visual question answering is less than BLIP-2.

**Strengths:**

The motivation is clear and the writing can be easy to be followed.

The multimodal COT reasoning ability is analyzed which is interesting.

The pretrained multimodal large language model achieve impressive zero-shot performance on many downstream tasks.

**Weaknesses:**

This paper presents a lot of experiments to demonstrate how good their model is, but it does not compare GPT-4 or BLIP-2 to show how much room for improvement their model has. Directly adopting a good image captioning model to convert images into diverse text and feeding them into a large language model (gpt-3.5 or llama) will also achieve good results (please see the multimodal instruction-following data construction way shown in the paper ``visual instruction tuning’’ ). The technique novelty advantage of this model compared to the common approach should be elaborated.

This paper claims that they proposed a multi-modal large language model that can perceive general modalities, learn in context (i.e., few-shot), and follow instructions (i.e., zero-shot), but the experimental baseline always lacks some key models that are relatively strong in the field, such as the experiment results (Table 6) shown in Section 3.8: language tasks. They only compared the language model trained on their own corpus. In fact, the multimodal version of KOSMOS-1 sees more text data and image description, leading to the comparison a little unfair. On the one hand, the comparison is not sufficient because there is no comparison with other large language models. What are the overall advantages and core contribution of KOSMOS-1 compared to LLMs (such as LLaMA, BLOOM, OPT) and previous multimodal models (BLIP-2, Flamingo)?

In addition, Sec 3.6 Multimodal Chain-of-Thought Prompting analyzed the multimodal COT reasoning ability, but it is only evaluated on the classification task SST-2. There is no demonstration of corresponding examples and no evaluation on complex reasoning tasks (e.g., ScienceQA), so it is difficult to judge whether this KOSMOS-1 really has the multimodal COT reasoning ability. Because prior work has shown that the ability of the chain of thought may emerge when the number of parameters of large language model is greater than 10B. The conclusion drawn by this paper is hard to convince.

Finally, Sec 2.4:Training Objective does not specify in detail how the visual representation model and language model are trained. If you train separately, how do you perform multimodal alignment and fusion? From the description, what is the difference between this whole training process and Oscar's training method? If following your description to train KOSMOS-1, will the model have the multi-modal COT reasoning ability?

In sec.2.1, should the input textual token embedding and visual token embedding be pre-aligned and then feed to the decoder? Or is it entirely dependent on the learning ability of the following decoder.

**Questions:**

See the weaknesses part.

---

> ### Author Rebuttal · Authors · 2023-08-10
>
> Thank you for your valuable feedback and suggestions on our paper.
>
> **About compared to LLM**
> > They only compared the language model trained on their own corpus. In fact, the multimodal version of KOSMOS-1 sees more text data and image description, leading to the comparison a little unfair. On the one hand, the comparison is not sufficient because there is no comparison with other large language models.
>
> For fair comparisons, we train the language model baseline with the same text corpora and training setup. They have the same number of training tokens.
>
> Other LLMs use different architectures, data, hyperparameters, steps, etc., resulting in too many incomparable factors. Therefore, we specifically align the settings for a fair comparison, allowing for more scientific comparisons.
>
> **About core contribution compared to LLMs and previous multimodal models**
> > What are the overall advantages and core contribution of KOSMOS-1 compared to LLMs (such as LLaMA, BLOOM, OPT) and previous multimodal models (BLIP-2, Flamingo)?
>
> Compared to LLMs:
> 1) We are a Multimodal Large Language Model (MLLM), supporting multimodal input and cross-modality transfer.
> 2) Our model brings more possibilities and application scenarios, such as embodied AI.
> 3) Moreover, our training data is different from LLMs, which use pure text corpora. We employ datasets like text corpora, image-caption pairs, and interleaved image-text data.
>
> Compared to previous multimodal models:
> 1) We train our model from scratch, enabling it to learn visual commonsense through cross-modality transfer.
> 2) We can perform in-context learning, while BLIP-2 lacks the ability to perform in-context learning (as stated in their Limitation section)[1].
> 3) Although our model size is relatively small, the results are impressive, indicating great potential.
> 4) In terms of the methodology, our model implementation is remarkably simple, following a minimalist approach.
> 5) In terms of evaluation, we explore OCR-free language understanding, text instruction tuning, Raven IQ tests, customized image classifiers, and cross-modality transfer, which have not been analyzed in previous works.
>
> [1] Li, Junnan, et al. "Blip-2: Bootstrapping language-image pre-training with frozen image encoders and large language models." arXiv preprint arXiv:2301.12597 (2023).
>
> **No evaluation on complex reasoning tasks (e.g., ScienceQA)**
> > There is no demonstration of corresponding examples and no evaluation on complex reasoning tasks (e.g., ScienceQA), so it is difficult to judge whether this KOSMOS-1 really has the multimodal COT reasoning ability. Because prior work has shown that the ability of the chain of thought may emerge when the number of parameters of large language model is greater than 10B.
>
> We evaluate KOSMOS-1 on ScienceQA benchmark under zero-shot and few-shot settings with and without explanations. The table below presents the results on ScienceQA test image set. We find that chain-of-thought prompting via introducing explanations performs slightly better than standard prompting (Few-shot (k=1) in the Table), which demonstrates the chain-of-thought ability of KOSMOS-1 on complex reasoning tasks. We believe that chain-of-thought prompting can bring greater improvements as the model size increases, as the text part improves.
>
> | Setting   | ScienceQA |
> |-----------|----------------|
> | Zero-shot                      | 56.1    |
> | Few-shot (k=1)                 | 56.8   |
> | Few-shot (k=1) w/ explanations | 57.2  |
>
> **About training objective**
> > Finally, Sec 2.4:Training Objective does not specify in detail how the visual representation model and language model are trained. If you train separately, how do you perform multimodal alignment and fusion? From the description, what is the difference between this whole training process and Oscar’s training method?
>
> We train the language decoder and visual encoder together via the next-token prediction task on text corpora, image-text pairs and interleaved image-text data.
>
> Given the input containing images and texts, we obtain image embeddings and text embeddings via the visual encoder and lookup table of word embeddings. Then we feed these image and text embeddings into the Transformer-based decoder. The self-attention module in the decoder can fuse the images and texts.
> For the training on image-text pairs and interleaved data, the model is trained to generate captions/texts based on images, which helps the model to learn its alignment.
>
> Compared to Oscar:
> 1) The main component of KOSMOS-1 is a Transformer-based decoder, which is unidirectional, processing the input sequence only from left to right. While Oscar is a bidirectional Transformer-based encoder.
> 2) KOSMOS-1 shows zero-shot and in-context learning capabilities on vision-language tasks, Oscar is mainly used for fine-tuning on specific vision-language tasks.
> 3) KOSMOS-1 is trained using next-token prediction task, while Oscar is trained using masked token predication since it is a bidirectional model.
>
> **About CoT reasoning ability**
> > If following your description to train KOSMOS-1, will the model have the multi-modal COT reasoning ability?
>
> CoT (or step-by-step generation) ability can be inherited from the capabilities of text language models, and we will also conduct more experiments on a larger scale.
>
> **About pre-align textual token embedding and visual token embedding**
> > In sec.2.1, should the input textual token embedding and visual token embedding be pre-aligned and then feed to the decoder? Or is it entirely dependent on the learning ability of the following decoder.
>
> It is not necessary to pre-align the input textual token embeddings and visual token embeddings before feeding them to the decoder.
>
> We feed the visual and textual embeddings into the decoder, and train the model to generate the next token depending on the previous context (images and texts). The training helps the model to align image and text representations.

---

> > ### Comment · Reviewer_BRko · 2023-08-22
> > **Official comments from BRko**
> >
> > I have read the authors' rebuttal as well as the other reviewers' comments. First the authors have addressed my comments as well other reviewers' comments. Second, the proposed method does have merits, which can use different data ( text corpora, image-text pairs and interleaved image-text data), and good model performances.
> >
> > Therefore, I am increasing my rating from Borderline Accept to Weak Accept.

---

### Official Review · Reviewer_XnP9 · 2023-06-27

**Soundness:** 3 good
**Presentation:** 3 good
**Contribution:** 2 fair
**Rating:** 6
**Confidence:** 4

**Summary:**

This work introduces KOSMOS-1, a multimodal large language model trained on large-scale text corpora, image-caption pairs, and interleaved image-text data. KOSMOS-1 can perform classic captioning/VQA tasks in a zero-shot or few-shot in-context prompting fashion. It can also perform OCR from visual document, and answer questions based on a webpage. It consists of a pre-trained CLIP-L/14 model for image representation and MAGNETO for language decoding. KOSMOS achieves competitive results on a wide suite of vision-language tasks with 1.6B parameters compared against Flamingo-9B. Furthermore, it introduces a Raven’s IQ benchmark to evaluate nonverbal intelligence.

**Strengths:**

The paper presents comprehensive technical details about the proposed system, including pre-trained models, loss design, and datasets. It also shows impressive qualitative sample usage of the proposed system on a wide range of tasks in supplemental, including multi-turn multimodal dialog, image classification with description, few-shot multimodal in-context learning, multimodal chain-of-thought prompting, and reasoning with webpage screenshots.

**Weaknesses:**

I do not find the proposed system particularly novel, because its architecture design, training losses, and pre-training datasets are widely adopted in prior art such as Flamingo [1] and BLIP-2 [2]. Furthermore, BLIP-2 [2] achieves stronger zero-shot performance on VQAv2 (Table 2b) 65% compared to KOSMOS-1's 51.0%, even though it is trained on much fewer data (LAION114M) with much fewer tunable parameters.

The small-scale Raven IQ test with 50 samples is undoubtedly challenging and interesting, however, it is hard to say KOSMOS-1 is better than random chance because it is only marginally better by 5.3%. Could authors discuss what are some promising directions to improve KOSMOS-1 on this benchmark?

Language-only instruction-following paradigm shows worse performance on COCO but leads to better results on Flickr30K, VQAv2, and VizWiz (Table 7). Could the author explain why language-only instruction-tuning might lead to better or worse results on different vision-language tasks?

To demonstrate that the KOSMOS-1 has better commonsense reasoning capabilities, the authors show language-only zero-shot evaluation results on RelativeSize, MemoryColor, and ColorTerms even though KOSMOS-1 is a multimodal model. Is there a reason not to report the multimodal zero-shot performance?

The authors do not promise a model release.

[1] Flamingo: a Visual Language Model for Few-Shot Learning. 2022.

[2] BLIP-2: Bootstrapping Language-Image Pre-training with Frozen Image Encoders and Large Language Models. 2023.

**Questions:**

- How can one improve KOSMOS-1 for better performance on Raven-IQ test?
- Why is the zero-shot VQAv2 performance of KOSMOS-1 worse than BLIP-2?
- Could authors report KOSMOS-1 performance on commonsense reasoning tasks while using multimodal inputs?
- What is the intuition behind language-only instruction-tuning? Why could this benefit vision-language tasks?
- Will the code and model be released to the public?

The writing can be further improved. For example:
L24: “it is still struggling to natively use LLMs for multimodal data, such as image, and audio.” -> “it still struggles to natively use LLMs for multimodal data, such as image and audio.”

**Limitations:**

Yes, the authors discuss limitation in appendix.

---

> ### Author Rebuttal · Authors · 2023-08-10
>
> Thank you for your valuable feedback and suggestions on our paper.
>
> **Improving Raven IQ test**
>
> > The small-scale Raven IQ test with 50 samples is undoubtedly challenging and interesting, however, it is hard to say KOSMOS-1 is better than random chance because it is only marginally better by 5.3%. Could authors discuss what are some promising directions to improve KOSMOS-1 on this benchmark?
>
> The performance gain looks marginal because the baseline is relatively low at 16.7%. The relative gain of KOSMOS-1 over the random chance is 31.8%, which looks more substantial.
> These are a few promising directions to explore for improving KOSMOS-1 on Raven IQ test:
> 1) **Increasing model capacity**: Scaling up the model size can potentially improve performance. This allows the model to learn more complex representations and better capture the relationships between different elements in the Raven IQ test task.
> 2) **Training data**: Enhancing the training data by incorporating more diverse and complex Raven-style problems can lead to better generalization.
> 3) **More fine-grained image representations**: Raven IQ test often requires a more fine-grained representation of images, which allows the model to understand the global relationships between multiple input images.
>
> **About Language-only Instruction-tuning**
> > Language-only instruction-following paradigm shows worse performance on COCO but leads to better results on Flickr30K, VQAv2, and VizWiz (Table 7). Could the author explain why language-only instruction-tuning might lead to better or worse results on different vision-language tasks?
>
> For image captioning tasks, we conduct some statistics on the COCO and Flickr30k datasets, and we find that the Flickr30k dataset has a higher average number of caption tokens. The model tends to generate longer outputs after instruction tuning, which might lead to some improvement in the Flickr30k dataset, but some decline in the COCO dataset.
>
> VQAv2 and VizWiz are question-answering tasks, where the model needs to understand both the question and the image content to provide an accurate answer. Language-only instruction-tuning tends to obtain better results on these tasks as the text-based instructions can help the model better follow instructions and improve its ability to answer questions.
>
> **Question 1**
> > Why is the zero-shot VQAv2 performance of KOSMOS-1 worse than BLIP-2?
>
> The zero-shot VQAv2 performance of KOSMOS-1 is not worse than BLIP-2.
> As shown in Table 2 of the BLIP-2 paper, the VQAv2 (test-dev) performance of the KOSMOS-1 model (1.3B text decoder + 300M image decoder) is 51.0 in the zero-shot setting, while the performance of BLIP-2 (2.7B text decoder + 300M image decoder) is 49.7. Given that the text decoder of KOSMOS-1 is smaller than that of BLIP-2, this demonstrates the superior performance of KOSMOS-1 in this context.
> BLIP-2 achieves a VQAv2 score of 65.0 in the zero-shot setting when using the CLIP ViT-g (1.2B) image encoder and FlanT5XXL (11B) text decoder. We think that the significant improvement in performance can be attributed to the substantially larger model size.
>
> Additionally, the training dataset may have played a role in the performance disparity. BLIP-2 included the COCO dataset in their training, which is the source of images for VQAv2. Consequently, the model can learn in-domain knowledge from COCO dataset, further enhancing its performance on VQAv2 tasks.
>
> **Question 2**
> > Could authors report KOSMOS-1 performance on commonsense reasoning tasks while using multimodal inputs?
> To demonstrate that the KOSMOS-1 has better commonsense reasoning capabilities, the authors show language-only zero-shot evaluation results on RelativeSize, MemoryColor, and ColorTerms even though KOSMOS-1 is a multimodal model. Is there a reason not to report the multimodal zero-shot performance?
>
> We have conducted experiments on commonsense reasoning tasks while using multimodal inputs. As the original dataset does not provide images of objects, we obtained relevant images using Google Image Search. These images were then prepended to the text descriptions and fed into the model. The results of these experiments are presented in the following table:
>
> | Model                                                 | RelativeSize | MemoryColor | ColorTerms |
> |------------------------------------------|--------------|-------------|------------|
> | KOSMOS-1                                          | 94.2         | 76.1        | 73.1       |
> | KOSMOS-1 + Google searched image | 80.7         | 82.6        | 86.5       |
>
> The table reveals that KOSMOS-1's performance on object color tasks (MemoryColor and ColorTerms) is improved when using multimodal inputs.
> We found that Google searched images are relatively noisy, which can affect the predictions. As a result, the performance of the RelativeSize task may decline.  In this case, it is better to use text for prediction directly, which also demonstrates the importance of our model's cross-modality.
>
> **Question 3**
> > What is the intuition behind language-only instruction-tuning? Why could this benefit vision-language tasks?
>
> The motivation of language-only instruction-tuning is to explore cross-modal transfer in KOSMOS-1 whether a model uses information learned in one modality (e.g., language) to enhance its performance in another modality (e.g., vision). As shown in Table 7, language-only instruction-tuning helps the model follow instructions. It helps the model to better understand the questions and generate answers in an appropriate format.
>
> **Question 4**
> > Will the code and model be released to the public?
>
> Yes, we will indeed release our model and code to the public.

---

> > ### Comment · Reviewer_XnP9 · 2023-08-20
> > **The authors have addressed my concerns**
> >
> > I will retain my positive rating as the authors promised to release the model and dataset and have addressed all of my concerns.

---

### Official Review · Reviewer_oWBZ · 2023-07-07

**Soundness:** 3 good
**Presentation:** 4 excellent
**Contribution:** 3 good
**Rating:** 6
**Confidence:** 4

**Summary:**

This paper presents a vision/language model trained on text and interleaved image/text data. It uses a ViT to encode the image into tokens and a transformer predict output tokens from the previous ones. It is trained on web-scale data and then fine-tuned on NLP instruction tuning data.

**Strengths:**

- Despite a lot of interest in instruction following for V/L models, training them from scratch is a somewhat less well explored area so I think contributions to this direction are useful.
- Many evaluations that help understand how effective the model is and what it has learned in different settings
- Showing NLP instruction tuning benefits multi-modal tasks is very interesting, although I would have liked see the ablation consider tasks with more complex instruction then just captioning/VQA kind of tasks. I also found the the chain-of-thought and visual common sense results interesting.


**Weaknesses:**

- The models appears to be limited in its ability to take advantage of few-shot data, which I think should be one of the main theoretical advantage of supporting interleaved V/L data and pre-training on such data. Captioning seems to be the only task where we really see a clear benefit.
- The instruction following shown in the paper is a bit simple, just for tasks like captioning, QA or classification which are tasks that are pretty close to the pre-training tasks. While it is interesting that this data helps the model. it has not really been shown the model can follow instruction for very different tasks in the same way LLM can.
- Related work is poorly discussed. There is no related work section, and as far I can see there is not much discussion about similar works such as other V/L foundation models or models like BLIP that adapt a pre-trained LLM in other parts of the paper or appendix either.

Overall I feel like the model and many experiments have scientific value despite the relatively smaller scale compared to other recent models, but I also feel like the lack related work is a non-trivial issue and I am not really sure how to balance those two points. I have recommend accepting the paper for now since I still think the paper would a benefit for the conference.

**Questions:**

Will the model or the interleaved data corpus be released?

Is there any evidence the model "forgets" any of its multi-modal knowledge during instruction fine-tuning, which only has NLP data? Is this a concern?

Did the authors consider initializing the model somehow with a pre-trained LLM? With so much work following that approach recently it would be interesting to hear why everything was trained from scratch.

**Limitations:**

Yes.

---

> ### Author Rebuttal · Authors · 2023-08-10
>
> Thank you for your valuable feedback and suggestions on our paper.
>
> **About related work**
> > Related work is poorly discussed. There is no related work section, and as far I can see there is not much discussion about similar works such as other V/L foundation models or models like BLIP that adapt a pre-trained LLM in other parts of the paper or appendix either.
>
> We will incorporate a discussion of related work in the final version.
>
> **Question 1**
> > Will the model or the interleaved data corpus be released?
>
> We will indeed release our model for the benefit of the research community.
> Without violating copyright restrictions and compliance, we will release the interleaved data corpus.
>
> **Question 2**
> > Is there any evidence the model “forgets” any of its multi-modal knowledge during instruction fine-tuning, which only has NLP data? Is this a concern?
>
> An ablation study was conducted to assess the impact of language-only instruction tuning on the model's performance (Table 7). The results indicate that language-only instruction tuning leads to improved performance on VL tasks.
>
> **Question 3**
> >Did the authors consider initializing the model somehow with a pre-trained LLM? With so much work following that approach recently it would be interesting to hear why everything was trained from scratch.
>
> Here are some key factors to this decision:
> 1) **Cross-modality transfer**: One of our goals is to achieve cross-modality transfer and enable language models to learn multimodal commonsense. By training from scratch, the model can be designed specifically to handle such transfers.
> 2) **Generalization through large-scale training**: Large-scale training tends to result in better generalization, as the model is exposed to more diverse data. Starting from scratch allows for the incorporation of more diverse data during the training process.
> 3) **Native multimodal LLM**: The model is a native multimodal LLM, meaning it has been designed from the ground up with multimodal learning in mind.
> 4) **Results drop in small size model**: We conducted a comprehensive comparison between training from scratch and initializing with a pre-trained LLM (text decoder contains 110 million parameters, and the image encoder is initialized from CLIP ViT-B/16. The training step is 300k). For a fair comparison, we trained these two settings with the same number of training steps. The results, as shown in the table below, indicate a decline in performance when initializing with a small-sized pre-trained LLM.
>
>  | Model           | COCO (CIDEr) | Flickr30k (CIDEr) | VQAv2 (VQA-acc) |
> |----------        |------- |--------------|-------|
> | From scratch     | 75.2  | 59.1            | 37.0  |
> | Cont. training   | 67.6  | 51.3            | 35.1  |
>
> Further research by PaLM-E [1] corroborated these findings (Figure 6), revealing that concatenating two pre-trained models could lead to a substantial performance degradation on NLP tasks, when the model size is small. To avoid this issue, we chose to train the model from scratch.
>
> [1] Driess, Danny, et al. "PaLM-E: An embodied multimodal language model." arXiv preprint arXiv:2303.03378 (2023).

---

> > ### Comment · Reviewer_oWBZ · 2023-08-21
> >
> > Thank you for answering my questions.

---

### Official Review · Reviewer_tesr · 2023-07-11

**Soundness:** 3 good
**Presentation:** 3 good
**Contribution:** 3 good
**Rating:** 6
**Confidence:** 5

**Summary:**

This paper proposed KOSMOS-1, a Multimodal Large Language Model (MLLM) that can take image embeddings as additional input to auto-regressive LLM. Trained on both text-only and image-text web-crawled data, KOSMOS-1 can get strong performance across a variety of tasks including language understanding, perception-language tasks, and vision tasks, demonstrating the utility of cross-modal knowledge transfer.

**Strengths:**

1. Although there are no specific novelties in terms of loss and module design, the introduced model is simple, unified, versatile, and effective.
2. The experiments cover both language-focused, vision-language, and vision-focused evaluations and demonstrate the strong performance of the proposed models.
3. The study about whether one modality can benefit the other (cross-modal transfer) is interesting and showcase some insightful observations.

**Weaknesses:**

1. Comparison with recent MLLM works either in related work or experiments was missing. For example, MiniGPT4, LLaVA, etc.
2. Although in ablation about cross-modal transfer, authors show that Language-Only Instruction Tuning can help several visual-language tasks. However, that experiment is only about language instruction tuning data. I wonder if the general Text Corpora (not just language instruction tuning data) in pre-training help general vision-language downstream tasks?
3. Can Visual Instruction Tuning data (LLAVA's or MiniGPT4's) help NLP tasks?

**Questions:**

See weaknesses.

**Limitations:**

Discussed in Supp.

---

> ### Author Rebuttal · Authors · 2023-08-10
>
> We thank the reviewer for the insightful comments.
>
> **Question 1**
> > Comparison with recent MLLM works either in related work or experiments was missing. For example, MiniGPT4, LLaVA, etc.
>
> We will add the comparison with recent MLLM works. The table below presents the comparison with MiniGPT4 and LLaVA on zero-shot image captioning (COCO and Flickr30k) and visual question answering (VQAv2). Since MiniGPT4 and LLaVA do not report results on these benchmarks, we test their released model on them ourselves. We use the MiniGPT4 version 'Vicuna-7B' and the LLaVA version 'LLaVA-Lightning-MPT-7B-preview'.
>
> | Model          | Model Size  | COCO (CIDEr) | Flickr30k (CIDEr) | VQAv2 (VQA-acc) |
> |----------        |-------  |------- |--------------|-------|
> | KOSMOS-1 | 1.6B | 84.7  | 67.1            | 51.0  |
> | MiniGPT4   | 7.3B | 86.3  | 54.4            | 28.9  |
> | LLaVA          | 7.3B | 74.3  | 45.9           | 34.9  |
>
> Experimental results show that KOSMOS-1 outperforms both MiniGPT4 and LLaVA in terms of Flickr30k and VQAv2 tasks. Our model also achieves competitive performance on COCO captioning.
>
> **Question 2**
> > Although in ablation about cross-modal transfer, authors show that Language-Only Instruction Tuning can help several visual-language tasks. However, that experiment is only about language instruction tuning data. I wonder if the general Text Corpora (not just language instruction tuning data) in pre-training help general vision-language downstream tasks?
>
> We conducted an ablation study on the general text corpora in a smaller setting, where the text decoder contains 300 million parameters, and the image encoder is initialized from CLIP ViT-B/16. The training step is 100k.
> The results are presented in the accompanying table. The general text corpora improve the performance on visual question answering (VQAv2) tasks, while resulting in a decline in the model's performance on image captioning (COCO, Flickr30k). The downstream COCO and Flickr30k image captioning data are similar to the image-caption pairs used in training (LAION-2B and COYO-700M). Adding general text corpora into training prevents the model from converging towards captioning tasks and results in a slight drop. But training on general text corpora improves the model to follow instructions. Learning from text question answering data improves the model to achieve a better performance on VQAv2.
>
> | Dataset                                                          | COCO (CIDEr) | Flickr30k (CIDEr) | VQAv2 (VQA-acc) |
> |------------------------------------------------------------------|------|-----------|-------|
> | Text Corpora + Image-Capton Pairs  + Interleaved Image-Text Data | 74.9 | 52.5      | 33.4  |
> | Image-Capton Pairs  + Interleaved Image-Text Data                | 77.8 | 54.1      | 28.1  |
>
> **Question 3**
> > Can Visual Instruction Tuning data (LLAVA's or MiniGPT4's) help NLP tasks?
>
> We perform instruction tuning with LLaVA’s visual instruction tuning data and evaluate the model on NLP tasks. As shown in the table, performing visual instruction tuning improves the zero-shot performance on HellaSwag, Winograd, Winogrande, PIQA and COPA. But we also observe a decline on StoryCloze and a large drop on BoolQ. Overall, introducing visual instruction tuning data does not significantly help NLP tasks, and we will explore more in the future.
> In addition, we evaluate the model on vision-language tasks and find that adding visual instruction data significantly improves the zero-shot performance on Flickr30k and VQAv2.
>
> | Model             | StoryCloze | HellaSwag | Winograd | Winogrande | PIQA | BoolQ | COPA |
> |-------------------|------------|-----------|----------|------------|------|-------|------|
> | KOSMOS-1          | 72.1       | 50.0      | 69.8     | 54.8       | 72.9 | 56.4  | 63.0 |
> | + Visual instruction tunning | 71.6       | 50.6      | 70.5     | 55.7       | 73.0 | 50.8  | 69.0 |

---

> ### Comment · Reviewer_tesr · 2023-08-21
>
> Thank the authors for adding the comparison and explanation. I am inclined to keep the rating.

---

### Decision · Program_Chairs · 2023-09-21

**Decision:**

Accept (poster)

**Comment:**

All reviewers were in favor of this paper pre-rebuttal, however there were serious concerns on: (i) the lack of novelty and insights (XnP9), (iii) the lack of a related works section (oWBZ), and (iii) missing comparisons to other LLMs and MLLMs (tesr, BRKo). That said, the reviewers liked the simplicity of the model and the notion of training the MLLM from scratch while demonstrating reasonable performance improvements against chosen baselines even with smaller model sizes. The authors presented a strong rebuttal with new ablation results and comparisons to other models. Given the authors have promised to publicly release their model, the reviewers have unanimously agreed to accept the paper.

The AC thinks that the empirical results in the paper are interesting and the public release of the pre-trained model will be valuable to the MLLM community; thus has decided to accept this paper. However, the AC recommends the authors to revise the camera-ready for: (i) better presentation of the paper, especially the Introduction, (ii) including a related works section, and (iii) improving the technical quality, e.g., via including relevant details of prior works that the paper builds on, such as MetaLM, Megneto, etc.